# Impacts of Climate Change and Remote Natural Catastrophes on EU Flood Insurance Markets: An Analysis of Soft and Hard Reinsurance Markets for Flood Coverage

**Max Tesselaar [1,](https://orcid.org)**, **W. J. Wouter Botzen [1,2,3]** and **Jeroen C.J.H. Aerts [1,4]**

[1] Institute for Environmental Studies, VU University Amsterdam, De Boelelaan 1087, 1081 HV Amsterdam, The Netherlands; wouter.botzen@vu.nl (W.J.W.B.); jeroen.aerts@vu.nl (J.C.J.H.A.)

[2] Utrecht University School of Economics, Utrecht University, Kriekenpitplein 21-22, 3584 EC Utrecht, The Netherlands

[3] Risk Management and Decision Processes Center, The Wharton School, University of Pennsylvania, 3819 Chestnut Street, Philadelphia, PA 19104-5340, USA

[4] Deltares, Boussinesqweg 1, 2629 HV Delft, The Netherlands

[*] Correspondence: m.tesselaar@vu.nl; Tel.: +31-621904562

**Abstract:** The increasing frequency and severity of natural catastrophes due to climate change is expected to cause higher natural disaster losses in the future. Reinsurance companies bear a large share of this risk in the form of excess-of-loss coverage, where they underwrite the most extreme portion of insurers' risk portfolios. Past experience has shown that after a very large natural disaster, or multiple disasters in close succession, the recapitalization need of reinsurers could trigger a "hard" reinsurance capital market, where a high demand for capital increases the price charged by investors, which is opposed to a "soft" market, where there is a high availability of capital for reinsurers. Consequently, the rising costs of underwriting are transferred to insurers, which ultimately could trigger higher premiums for natural catastrophe (NatCat) insurance worldwide. Here, we study the vulnerability of riverine flood insurance systems in the EU to global reinsurance market conditions and climate change. To do so, we apply the "Dynamic Integrated Flood Insurance" (DIFI) model, and compare insurance premiums, unaffordability, and the uptake for soft and hard reinsurance market conditions under an average and extreme scenario of climate change. We find that a rising average and higher variance of flood risk towards the end of the century can increase flood insurance premiums and cause higher premium volatility resulting from global reinsurance market conditions. Under a "mild" scenario of climate change, the projected yearly premiums for EU countries, combined, are €1380 higher under a hard compared to a soft reinsurance capital market in 2080. For a high-end climate change scenario, this difference becomes €3220. The rise in premiums causes problems with the unaffordability of flood coverage and results in a declining demand for flood insurance, which increases the financial vulnerability of households to flooding. A proposed solution is to introduce government reinsurance for flood risk, as governments can often provide cheaper reinsurance coverage and are less subject to the volatility of the capital markets.

**Keywords:** climate change; flood (re)insurance; unaffordability; market penetration; capital markets; vulnerability; remote impacts

## 1. Introduction

Climate change will increase the frequency and severity of natural catastrophes [1], causing direct damage to built environments and agricultural land, as well as indirect economic impacts, through

trade- and supply-chain disruptions, lower productivity, and loss of income [2]. Furthermore, large natural catastrophes can have economic impacts on the financial market. For example, the demand for capital that is needed to rebuild assets after a natural catastrophe can cause the price of capital to rise, which can increase the cost of insurance as insurers are trying to recapitalize after large amounts of claims are paid [3]. The vulnerability of the insurance sector, due to remote catastrophes that occur outside the EU, can have directly-felt consequences for households in EU countries regarding coverage and costs, which will be the main focus of this study.

Insurance is a way to enhance financial resilience against damage caused by natural events, as it is found to increase the certainty of adequate financial compensation after an event and therefore increases the speed of recovery [4,5]. However, premiums for insurance against natural disasters are expected to rise in the future as a result of climate change [6], which can cause it to become unaffordable for certain population groups, or it can exceed the willingness-to-pay for insurance. This means that insurance uptake can decline in markets where purchasing flood coverage is optional. It is therefore important to assess under which conditions insurance premiums are affected by climate change and to understand how to limit potential increases in premiums.

The insurance industry aims at absorbing individual losses by spreading risk over time and space. This means that the risk of individual insurance companies becoming insolvent declines with the geographical spread of its policyholders, thereby creating a larger risk-pool with lower correlated risk. By calculating average risk over time, the insurance mechanism can transform a large financial shock due to a disastrous event into a manageable annual premium. As the primary insurer's ability to spread its risk is often limited geographically by, for example, national borders, large-scale natural catastrophes can jeopardize the solvability to settle indemnity payments. Therefore, the insurer can choose to transfer part of its risk portfolio to a reinsurer. An often-used mechanism for this is excess-of-loss coverage, where reinsurance coverage steps in when a certain threshold of insurance claims has been made. Therefore, the reinsurer deals with the most extreme risks, which are events that occur with low probability but have a high impact. In order to maintain a manageable risk portfolio the reinsurer often operates on a global level, and due to scale-advantages on this level, the market is dominated by few firms compared to the market for primary insurance [7]. As a result, there is limited competition in this market, which can cause reinsurance premiums to be several times the actuarial price of the underwritten risk [8].

The reinsurance premiums generally fluctuate due to underwriting cycles, which are periods of rising and falling underwriting profits as a result of fluctuating prices of coverage [3]. Internal forces of supply and demand cause these fluctuations, where periods of increased competition between reinsurers and products leads to declining premiums and an expansion of coverage. Due to lower returns on the underwritten risk and declining profitability, some (re)insurers will withdraw from the market, at which point competition decreases and premiums rise. When profitability is restored, new insurers and products will enter the market, and the cycle starts over again [9].

In addition to inherent cycles in the market for risk financing, price fluctuations can arise due to external events such as large natural- or man-made catastrophes, which may cause a reinsurer's capital stock to deteriorate. Consequently, the demand for capital increases, resulting in higher prices demanded by investors and, ultimately, rising reinsurance premiums [7,8]. Resulting from an increase in the return on capital, however, the supply of capital increases, causing the reinsurance premium to decline again. Therefore, a determining factor for the reinsurance premiums is the availability of capital. When capital is scarce as a result of large-scale demand after a natural- or man-made catastrophe, market forces lead to what is referred to as a "hard" market for capital, where the price of capital rises [10]. Reinsurance premiums can decline again in a "soft" market, which is a result of the increasing supply of risk coverage [11]. As reinsurers usually operate worldwide, and there are often multiple reinsurers engaged in an area affected by a natural disaster, these conditions can be transmitted to primary insurers across the globe [12]. Past experience supports that, as a result of rising reinsurance costs in a hard capital market, premiums for natural disaster insurance can increase. The

OECD (2018) [13] found a high correlation (79%) between property catastrophe reinsurance pricing and rates of primary insurance against natural hazards in the US.

Looking at further empirical evidence of this relationship, 2005 was a very costly year for property insurance due to three hurricanes that made landfall in the US (Katrina, Rita and Wilma with a combined loss of $79 billion) [14]. As a result, reinsurers were forced to recapitalize and the Global Rate on Line (RoL) Index, an annual measure of change in property catastrophe reinsurance premiums, increased by 36.6% [15]. The following two years had no major natural disasters and, as a result, the capital of reinsurers is largely replenished and reinsurance premiums are in decline [13]. The next rise in reinsurance premiums resulting from a natural disaster occurs in 2011, mainly as a result of the Tohoku earthquake ($40 billion insured losses [16]) and consequent tsunami in Japan, which led to an increase in reinsurance premiums of 9.5% [15]. These examples, from practice, show how hard reinsurance markets did arise in the past, despite the practice of reinsurers to spread various types of risks, including both catastrophe and non-catastrophe risk portfolios. For example, a reinsurer's risk portfolio can, besides the risk of natural hazards, also contain underwritten life- and health-risks. In this study, we emphasize the impact of natural disasters on capital market volatility for reinsurers, as the risk of natural disasters is found to be increasing as a result of climate change. However, a hard capital market for reinsurance could also be triggered by other events, such as an outbreak of an epidemic disease. It is noteworthy that we do not distinguish between the types of disasters and events that trigger a hard capital market and hence do not consider risk spreading over different types of risk portfolios. Instead, we examine the implications of hard market conditions arising in the future, as they have occurred in the past, for flood insurance premiums and other relevant indicators of the performance of flood insurance markets.

This study progresses on the previous study by Tesselaar et al. [17] by investigating the impact of soft and hard reinsurance markets for natural disasters, in addition to the impact of climate change, on flood insurance in the EU. The frequency and severity of natural catastrophes is increasing worldwide, which can lead to more occurrences of a hard market in the future, causing insurance premiums within the EU to be higher than expected. Rising insurance premiums for natural hazards, such as flooding, increase the basic living expenses of households, which may reduce a certain standard of living after obtaining insurance coverage. As shown by Tesselaar et al. [17], who projected the impact of climate- and socio-economic change on flood insurance premiums in the EU, insurance can become unaffordable for lower income groups. Also, when natural hazard insurance is optional, rising premiums may discourage individuals from obtaining coverage, leaving them more financially vulnerable to the impact of natural disasters [18]. Therefore, we aim to show how conditions for the EU flood insurance market are dependent on global disasters, and what the possible outcomes are in terms of the premiums, unaffordability, and uptake of flood insurance for private households. By using the "Dynamic Integrated Flood Insurance" (DIFI) partial equilibrium model [19], we estimate flood insurance premiums for different stylized flood insurance systems under various scenarios of climate change, and we project the impact of a hard or soft reinsurance market on this outcome. Most EU countries currently maintain a (semi-) voluntary system for flood insurance, where primary insurers obtain excess-of-loss coverage from private reinsurers and are therefore subject to developments on the international reinsurance and capital market [19]. In the case of a hard market, the cost loading factor of the reinsurer is higher than in a soft market, causing the insurer to charge a higher premium for coverage.

In the following section, we elaborate on the applied methodology, after which we present the results. Then, we discuss the implications of the findings and conclude the paper.

## 2. Methodology

The modeling exercise uses an adapted version of the "Dynamic Integrated Flood Insurance" (DIFI) model [19], which is a partial equilibrium model developed to assess flood insurance systems based on economic efficiency and equity. More specifically, the DIFI model combines a spatially explicit

flood risk model with an insurance sector and a behavioral model in order to project, among other variables, insurance premium development over time and the resulting unaffordability of, and demand for, insurance. Since, for the purpose of this study, we do not analyze the full spectrum of output delivered by the DIFI model, we limit the model description in this section to the most relevant parts of it, and refer to the original model paper by Hudson et al. [19]. For simplicity, in the model description presented here, we follow the flow of the model, which starts with a flood risk model, the output of which flows into an insurance sector model, which then impacts a consumer behavior model.

### 2.1. Flood Risk Model

We apply the DIFI model version 2.0 [17], of which the difference with version 1.0 developed by Hudson et al. [19] is that raw flood damage data is obtained from the global flood risk model "GLOFRIS" [20,21]. As is common in these models, flood risk in GLOFRIS is determined by combining flood hazard and the extent and probability of flooding, with the exposed built environment and its vulnerability. Flood hazard is determined as the inundation extent associated with a certain return period (e.g., expected to occur once in every 50 years), with a spatial resolution of 30″ × 30″. It is calculated for four time steps, 2010, 2030, 2050, and 2080, which are all determined by taking the means of 40 year intervals around these years, serving as a representative climate in that year. For 2010, the average is taken over the period 1960–1999, for which climate data is used from the EU-WATCH project [22]. For future periods, the climatic data is obtained from the ISIMIP project [23], which contains five "global circulation models" (GCMs) that are forced with "representative concentration pathways" (RCPs) to simulate the level of greenhouse gas emission in future scenarios [24]. In this study, we compare climatic effects of RCP4.5, which is a greenhouse gas emission scenario broadly in line with the 2 °C target of the Paris climate accord, with RCP8.5, which is an extreme scenario of greenhouse emissions, aligned with a future where a high dependency on fossil fuel remains.

The impact that a flood of a certain size can have on a society depends on the value of the built environment that is affected by it, the exposure, as well as how much of it will be destroyed, which is determined by its vulnerability. In GLOFRIS, the exposure is simulated by overlaying the derived inundation maps with a land use map that specifies the exposed urban areas and population. For the baseline, the urban density map is taken from the HYDE database [25], while the current and future population is obtained from the 2UP model [26]. Since this study concerns flood insurance, which is organized on a household level, we divide the exposed population by the nationally determined average household size, which is taken from Eurostat (Household composition statistics over 2008–2018. Last accessed on June 11th 2019). The economic value of the exposed assets is determined based on national GDP per capita in 2010 [27]. For future periods, the exposure to flooding is approximated by using "shared socio-economic pathways" (SSPs), which are potential storylines that simulate economic and population growth [28]. The growth of per capita GDP and population is taken from IIASA's SSP database, while the simulations of exposed urban assets are taken from Winsemius et al. [21].

The vulnerability of a built environment to various levels of inundation is modeled using global flood depth-damage functions, which are taken from Huizinga et al. [29]. After this, flood hazard is combined with the exposure and vulnerability, and the model can determine "expected annual damage" (EAD) for the various scenario combinations in the RCP-GCM-SSP-space. At this point, as is common practice, we take the average over the output of the five GCMs in order to reduce biases that may originate from these models [30]. For further steps, we also aggregate the flood risk to the NUTS2 level (The NUTS regions (nomenclature of territorial units for statistics) is a hierarchical system for dividing up economic territories of the EU, where NUTS2 are basic regions for the application of regional policies (Eurostat)).

We then assess whether a certain water-level can actually cause an inundation of land, by comparing the modeled water-levels per return period with protection standards in place, which are taken from the modeled layer of the FLOPROS database [31]. For example, we assume that when dike heights in a region are designed to withstand a maximum water-level with a return-period of

1/100 years, water-levels with a probability of 1/50 years can cause no damage, while a water-level associated with a 1/200 year return period will cause damage. Furthermore, we assume constant protection standards, which means that governments will raise dike-heights accordingly with the increasing risk. This also means that for areas where the flood risk will decrease due to a dryer climate, the protection standard will increase, as we assume that dike-heights will not be lowered in the future. Then, in order to calculate the probability and extent of the exceedance of protection standards, we fit a damage probability curve based on a power law function (A power law function takes the form: $L = \tau_1 + \tau_2 p^{\tau_3}$). Finally, we are able to determine EAD and its variance by running a Monte Carlo simulation, which draws random return periods and compares it to the protection standards.

For the following steps, in the DIFI model, we focus our analysis on households located in 1/100-year flood zones, meaning those households that face a flood probability of 1/100 or higher. Having access to affordable and sufficient flood insurance coverage is most relevant for these households that face a high risk of flooding. Hence, the 1/100-year flood zones are areas where the debate about flood insurance issues, such as unaffordability, uptake, and adaptation measures, is most prominent. For example, the Federal Emergency Management Agency (FEMA) in the US made flood insurance uptake mandatory in the 1/100 flood risk zone for households with mortgages from federal lending institutions [32]. While, in the UK, the 1/75 year flood zone is a demarcation for a prioritized risk-reduction effort by the government [33].

## 2.2. Insurance Sector Model

In this part of the DIFI model, insurance premiums are calculated based on the EAD determined in the previous section. By interpolating the EAD between the time-steps 2010, 2030, 2050, and 2080, we are able to obtain a range of 70 years of flood insurance premiums. There are several stylized versions of flood insurance systems in the EU, which we present in the Figure 1. However, as not all of these use a private reinsurer and are therefore not prone to shocks on the international reinsurance market, we limit our analysis to the specific insurance structures that do use a private reinsurer. We classify two insurance mechanisms where this is the case: the voluntary and semi-voluntary systems. The voluntary system maintains risk-based premiums and optional coverage, and is applied in Germany, Italy, and Poland, among others. The semi-voluntary system is similar, except coverage is a mortgage requirement, as is the case in Sweden, or included in general home insurance, as is the case in Finland. We will now first explain the premium setting rules of the insurer in the (semi-) voluntary market structure and, afterwards, describe in detail how reinsurance impacts this.

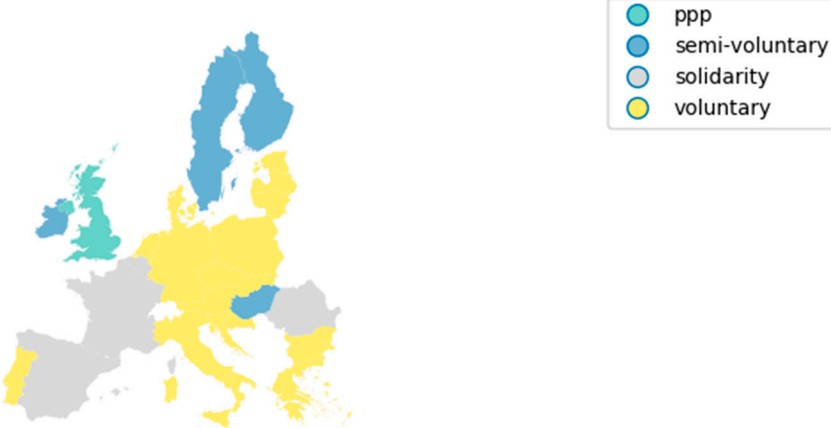

**Figure 1.** Stylized version of current flood insurance systems in EU countries. The voluntary system has risk-based premiums, private reinsurance, and a voluntary uptake; the semi-voluntary system is similar except uptake is required for a mortgage; the solidarity system has a mandatory uptake and no premium differentiation; the public–private partnership (ppp) has a mandatory uptake, limited risk-based premiums and a public reinsurer.

Equation (1) represents the process of calculating the (semi-) voluntary insurance premium, where $\overline{\pi}_{j,t}$ represents the risk-based premium for a 1/100-year floodplain on a NUTS2-level, as indicated by subscript $j$, and at time $t$. The first element of Equation (1) calculates the premium for the risk transferred to the insurer, whereas the second element determines the premium charged by the reinsurer for the risk it bears. The risk borne by the insurer is made up of ($\overline{L}_{j,t}$), representing the expected flood damage minus ($\overline{D}_{j,t}$), the deductible, which is set at 15% of the transferred loss. On top of the expected loss, the insurer charges for the uncertainty of damage, which is determined by multiplying a risk aversion coefficient $r$ with $\sigma_{0 < a < 99.8}$, the volatility of flood damage within a quantile range that is considered insurable [34]. The costs for the risk transferred to the insurer is multiplied by a loading factor $\dot{\lambda}_{c,t}$. For the primary insurer, we assume Bertrand competition, meaning that insurers compete over prices or loading factors, as a result of which the insurer can only charge a cost-loading factor and not a profit-loading factor. The cost-loading factor comprises the costs of providing insurance, such as administrative costs, and is determined as a fixed percentage of the insurance premium. For more details on how this is determined we refer to Hudson et al. [19].

The second element in Equation (1) expresses the premium for the risk transferred to the reinsurer. In the modelled (semi-) voluntary insurance market, the top 15% of the risk is transferred from the insurer to the reinsurer as excess-of-loss coverage. The reinsurer's cost of providing coverage is similar to the insurer's and is indicated here with the superscript *RR*. The main difference with the insurer's premium for this study is that we adjust the risk-aversion parameter $r$ for the reinsurer according to the reinsurance capital market conditions determined in Kunreuther et al. [11], which is $r = 0.4$ for a soft capital market and $r = 0.7$ for a hard capital market for reinsurance. These parameters are derived from stakeholder consultations that the authors conducted with (re)insurers, which makes these a useful approximation. However, the parameters are mainly indicative, meaning that actual market conditions can differ from these. Nevertheless, for the purpose of this study, these parameters provide a useful lower and upper bound of reinsurance premiums under soft and hard market conditions.

$$\overline{\pi}_{j,t} = \left(1 + \dot{\lambda}_{c,t}\right)\left(E\left(\overline{L}_{j,t}(p) - \overline{D}_{j,t}(p)\right) + r * \sigma_{0 < a < 99.8}\right) \\ + \left(1 + \ddot{\lambda}_{c,t}\right)\left(E\left(\overline{L^{RR}_{j,t}}(p) - \overline{D^{RR}_{j,t}}(p)\right) + r * \sigma^{RR}_{0 < a < 99.8}\right) \tag{1}$$

In the modeled version of the (semi-) voluntary insurance system, households are able to obtain a discount on the final premium if they implement risk-reduction measures such as dry- or wet-flood proofing. The decision to implement these measures by households is based on subjective expected utility maximization. We recognize that some household decisions may deviate from subjective expected utility maximization and may, for example, be influenced by the spread of risk. For a detailed description of the subjective expected utility maximization mechanism we refer to the original DIFI model article [19], which also includes an alternative individual decision theory as a sensitivity analysis. The final premium for households ($\pi_{i,j,t}$), as shown in Equation (2), is determined by multiplying the baseline average risk per household in a region ($\overline{\pi}_{j,t}$) by the risk after implementing the risk-reduction measures ($1 - ER_{DRR}$).

$$\pi_{i,j,t} = (1 - ER_{DRR})\overline{\pi}_{j,t} \tag{2}$$

### 2.3. Consumer Behavior Model

The insurance premiums, as estimated in the previous module, feed into a consumer behavior module, where households' decisions for purchasing insurance and implementing DRR measures are simulated.

The decision functions are based on subjective expected utility maximization [35], where, for each region, the decision process is iterated 10,000 times in order to limit the randomness bias. The model starts by determining how many households implement dry- and/or wet flood-proofing measures before being exposed to an insurance incentive. These decision functions are based on subjective flood risk, the perceived effectiveness of flood-proofing measures calibrated to survey data [36–38], and the

costs of the measure [39,40]. Based on whether a household implements DRR measures it can receive a discount on the insurance premium, as shown in Equation (2).

The household then chooses whether to purchase insurance, where it is firstly determined if insurance is affordable, and, if it is, the decision function is based on subjective expected utility maximization. Equation (3) summarizes the decision model for purchasing insurance, where the expected utility of insuring or not insuring is expressed in Equation (4). A household will buy insurance when the expected utility of insuring is higher and only when the premium charged ($\pi_{i,j,t,s}$) is equal or smaller than its poverty-adjusted disposable income (Income is drawn from a log-normal distribution, which is calibrated to regional mean and median income levels) $\left(Income_{i,j,t}\right)$ subtracted by the poverty line ($Poverty\ Line_{c,t}$), which is set at 60% of national median income.

$$U = \begin{cases} insure & if & E(U)_{1,i,j,t,s} < E(U)_{2,i,j,t,s} & s.t.\ \pi_{i,j,t,s} \leq Income_{i,j,t} - Poverty\ Line_{c,t} \\ not\ insure & if & E(U)_{1,i,j,t,s} \geq E(U)_{2,i,j,t,s} & or\ \ \pi_{i,j,t,s} > Income_{i,j,t} - Poverty\ Line_{c,t} \end{cases} \tag{3}$$

Equation (4) simulates which of $E\left(U_{1,i,j,t,s}\right)$ or $E\left(U_{2,i,j,t,s}\right)$ is highest, where the former considers the uninsured loss, and the latter the wealth ($W_{i,j,t}$) subtracted by the deductible ($0.15\gamma_{i,j}L_{i,j,t}(p)$) and the premium ($\pi_{i,j,t,s}$), with $\gamma_{i,j}$ being a parameter between 0 and 1 expressing households' flood risk misperceptions, which is calibrated based on data from the German Insurance Association [41]. The logarithmic functions express constant relative risk aversion, which is a common assumption for simulating household decision making under risk [42]. Furthermore, households consider flood losses over the probability range $[0, \widetilde{PS}_{i,j}]$, with $\widetilde{PS}_{i,j} = \vartheta_{i,j}PS_j$, and $\vartheta_{i,j}$ being a distribution of subjective flood occurrence probabilities (see the supplementary document of Hudson et al. [19] for more details).

$$E(U) = \begin{cases} E\left(U_{1,i,j,t,s}\right) = \int_0^{p=\widetilde{PS}_{i,j}} p\ln\left(W_{i,j,t} - \gamma_{i,j}L_{i,j,t}(p)\right)dp \\ E\left(U_{2,i,j,t,s}\right) = \int_0^{p=\widetilde{PS}_{i,j}} p\ln\left(W_{i,j,t} - 0.15\gamma_{i,j}L_{i,j,t}(p) - \pi_{i,j,t,s}\right)dp \end{cases} \tag{4}$$

After this, the insurance penetration rate, as a percentage of the population, and the level of unaffordability, can be estimated per NUTS2 region. Unaffordability can both be expressed as a percentage of the population for whom insurance is unaffordable based on our definition of unaffordability, as well as the absolute value of unaffordability, which is the regional cumulative of how much flood insurance premiums would cause households to fall below the poverty line. This is an important projection, as some policy proposals appeal for mandatory flood insurance [43], which may require funding support for households that cannot afford the premium [5].

## 3. Results

Here, we show the projected effects of reinsurance market volatility on flood insurance premiums, the unaffordability of insurance, and the market penetration rates. Unaffordability is shown as the absolute value of the unaffordable share of the premiums, as well as the percentage of the population for whom flood insurance is unaffordable. We show the output for two projections of climate- and socio-economic change: namely, RCP4.5-SSP2, which is roughly aligned with a 2-degree Celsius temperature rise and an average socio-economic development worldwide [44], as well as RCP8.5-SSP5, which is a high-end pathway for greenhouse gas emissions due to a high reliance on fossil fuels and high socio-economic growth [45].

Figure 2, below, shows projected insurance premiums for soft and hard reinsurance capital markets, on a national level, for 2020, 2050, and 2080. The premiums are determined based on the status-quo of national flood insurance systems, which is assumed to remain unchanged. Therefore, it can be seen that premiums in countries such as Belgium, Spain, France, and the UK are below average due to more cross-subsidization between high- and low-risk policyholders in their insurance systems (see Figure 1 for the allocation of insurance systems per country). For countries with (semi-) voluntary insurance systems, premiums are risk-reflective, which causes premiums in high-risk areas to be higher

than in areas with a lower risk. Since we show here the results for 1/100-year flood zones, it can be seen that premiums are projected to rise much more rapidly in these voluntary systems compared to countries where flood risk is born collectively. On average, premiums for countries with a solidarity insurance system are projected to rise with a factor 6.5 up to 2080 under hard conditions for RCP4.5, whereas this growth is a factor 9.3 for countries with a voluntary system. Under the RCP8.5 scenario, these growth rates are 15.5 and 24.4, respectively.

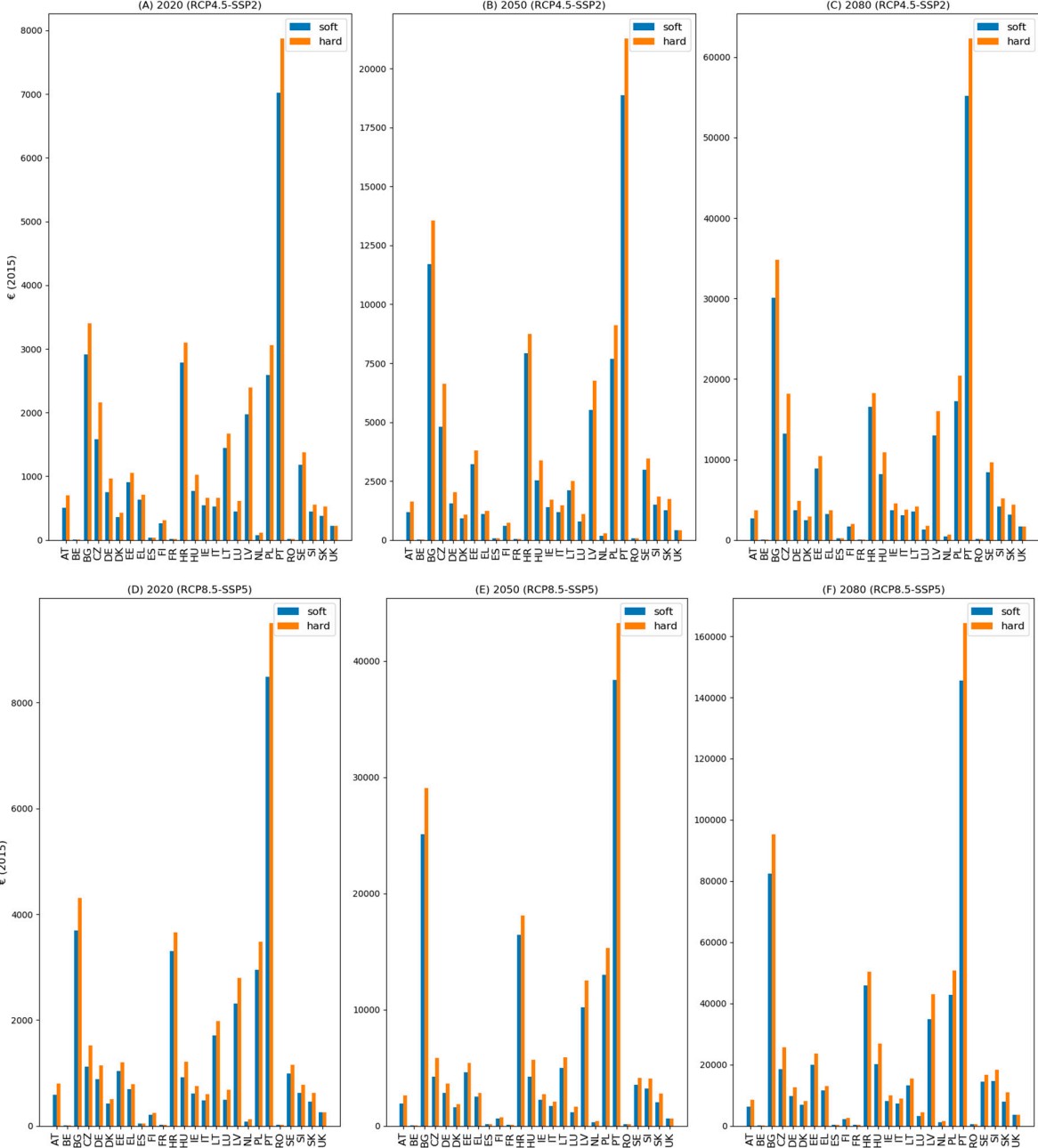

**Figure 2.** Average annual premiums in 2015 Euros for status-quo insurance systems in EU countries, for soft (blue) and hard (orange) reinsurance markets. Subplots **A**, **B** and **C** show results for 2020, 2050 and 2080, for RCP4.5-SSP2. Subplots **D**, **E** and **F** show results for 2020, 2030 and 2080, for RCP8.5-SSP5.

The premiums shown in Figure 2, are higher under the RCP8.5-SSP5 scenario (bottom panels) than under the RCP4.5-SSP2 scenario (top panels), which resonates the development of flood risk under the scenarios. Furthermore, it can be seen that premiums are consistently higher under a hard reinsurance

capital market than a soft version (shown in orange and blue colors, respectively). Although the exact ratio of the difference varies between countries, which is dependent on the variance of expected losses (see Equation (1)). For RCP4.5, the average difference in premiums between hard and soft markets for countries with a voluntary system rises from €154 in 2020 to €1710 in 2080. However, extreme differences in projections can be seen for Portugal (PT) and Bulgaria (BG), where the difference in premiums between market states for 2080 is €6500 and €4500, respectively. For RCP8.5, the average difference between capital market states for voluntary premiums rises to €3980 in 2080.

Figure 3 shows the impact of a hard and a soft capital market for reinsurance on the unaffordability of flood insurance for EU countries in 2020, 2050, and 2080. The graph shows the results for countries with (semi-) voluntary insurance arrangements, since these insurance systems are more prone to volatility on the capital market for reinsurers, as coverage is obtained from private reinsurers. Countries with a solidarity-based or a public–private partnership are excluded from later graphs, as these are not impacted by the capital market conditions resulting from remote natural catastrophes, because reinsurance is provided by governments.

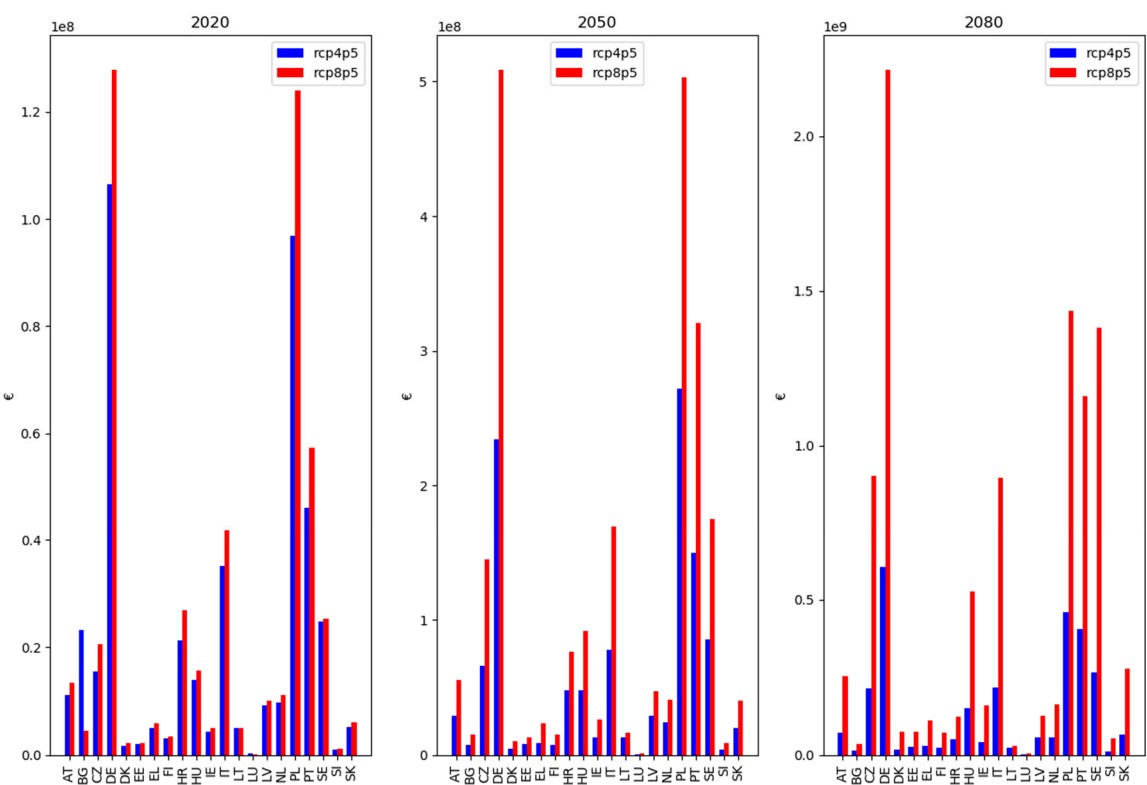

**Figure 3.** The difference in the total amount of unaffordability in Euros (2015) between hard and soft market conditions, shown for RCP4.5 (blue) and RCP8.5 (red), for individual EU countries in 2020, 2050, and 2080. Countries that provide public reinsurance for flood insurance are excluded here, as these are not exposed to global reinsurance market volatility resulting from large natural catastrophes.

Figure 3 shows the difference in the absolute value of unaffordability between a soft and a hard reinsurance market for both flood risk scenarios. For example, for 2020, it can be seen that for Germany (DE), under RCP4.5, unaffordability in a hard reinsurance market is slightly more than €100 million higher compared to a soft market. The large effect of reinsurance market volatility on insurance premiums in the second half of this century for RCP8.5, shown in Figure 2, can be seen to cause substantial differences in unaffordability when compared to RCP4.5. For example, the difference in the absolute value of unaffordability between soft and hard market conditions grows, on average, for EU countries, with a factor 5.2 for RCP4.5, while it grows with a factor 19.7 for RCP8.5. Figure 3 shows the sum of the unaffordable share of the premiums, which is an important indicator for national policies

that may be implemented to assist low-income households in obtaining flood insurance coverage. Due to a hard capital market, the average value of unaffordability in a country is approximately €15 million higher than for a soft market state in 2080 for RCP4.5, which, for RCP8.5, becomes €55 million. The largest difference in unaffordability between the two market states is expected in Germany (DE) in 2080 under RCP8.5, where the value of the unaffordable share of premiums for all households in high-risk areas is €2.2 billion higher under a hard compared to a soft reinsurance capital market. This result is not surprising, as Germany is the largest EU country in terms of population, and is expected to grow further towards 2080, which translates into a high absolute amount of unaffordable premium levels.

Figure 4 shows the unaffordability of flood insurance as a percentage of the population in high flood-risk zones. This indicates the share of the population that cannot afford flood insurance due to budgetary restrictions and which may require government assistance in order to obtain coverage. Unaffordability is shown for both soft and hard reinsurance market conditions in order to show the difference between the two market states, in addition to the extent of projected unaffordability as a result of climate- and socio-economic change. It can be seen that the percentage of the population that cannot afford flood insurance increases overall, as a result of climate- and socio-economic change. Under constant soft reinsurance market conditions, the percentage of households in high-risk areas for whom flood insurance is unaffordable rises from 23% in 2020 to 36% in 2080 for RCP4.5, while it rises to 41% under RCP8.5 (see Figure A1 in Appendix A). Several countries, including Croatia (HR), Latvia (LV), Poland (PL), and Portugal (PT), show significant increases over time. On average, over these countries, in soft market conditions, unaffordability rises from 30% in 2020 to 53% in 2080. The difference in reinsurance market conditions (the difference between the blue and red bars) shows consistently higher unaffordability under hard market conditions, with a slight variation between countries. Over all regions with (semi-) voluntary insurance, unaffordability is 2.4% higher in a hard compared to a soft capital market, which is relatively smaller than the difference in the absolute value of unaffordability shown in Figure 3. This is because most of the increased value of unaffordability in a hard market is a burden for households for whom insurance was already unaffordable in a soft market. In other words, the hard market adds to the total amount of unaffordability of premiums, but less to the share of the population who cannot buy insurance because it is unaffordable.

In Figure 4, it can be seen that the high absolute value of unaffordability in Germany, as shown in Figure 3, is shared amongst a much larger population than in, for example, Portugal, where the percentage of the population for whom flood insurance is deemed unaffordable is higher. In Portugal, for the RCP4.5 scenario, this projected percentage rises from 52% in 2020 under a soft market to 75% in 2080, which for a hard market becomes a rise from 54% to 78%. For several countries, the percentage of households for whom insurance is unaffordable remains low and changes only slightly (e.g., Austria (AT), Denmark (DK), the Netherlands (NL)), which is due to a combination of the low growth of riverine flood risk and the high projected income growth in these regions. The highest percentage of unaffordability is observed in Bulgaria (BG), where risk-based premiums are expected to be unaffordable for approximately 96% of the households located in high-risk areas in 2080 for both market states.

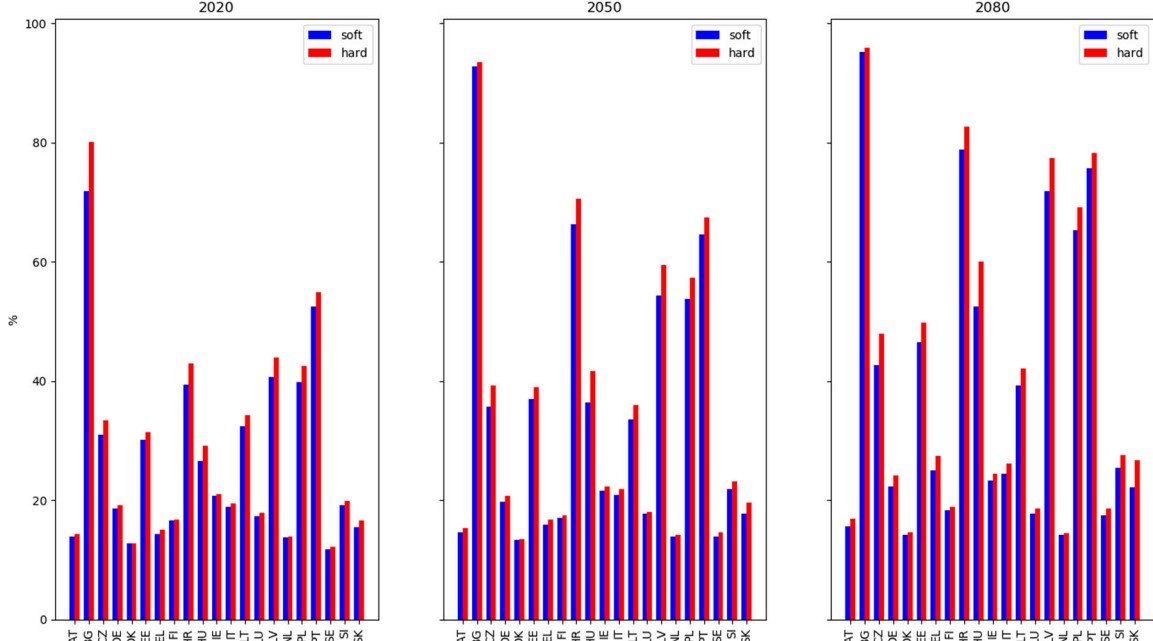

**Figure 4.** The percentage of the population that cannot afford insurance under soft market conditions (blue) and hard conditions (red) for EU countries, in 2020, 2050, and 2080, for RCP4.5. Countries that provide public reinsurance for flood insurance are excluded here, as these are not exposed to global reinsurance market volatility resulting from large natural catastrophes.

As affordability is a condition for insurance uptake, consistent with results shown in Figure 4, the projected insurance penetration rate in Bulgaria is low, as shown in Figure 5. In fact, the model estimations show a decline in insurance uptake from 12% in 2020 to merely 3% in 2080, for soft market conditions under RCP4.5. For other countries, while the projected uptake of insurance remains higher, a significant decline can be observed (e.g., Croatia (HR), Latvia (LV), Poland (PL), and Portugal (PT)), which is a result of increasing unaffordability, as well as a lacking willingness-to-pay for premiums that are higher due to climate- and socio-economic change or, as can be seen, due to hard reinsurance market conditions. For the five countries with the highest change in penetration rates, the uptake declines from 28% in 2020 to 13% in 2080 for soft market conditions under RCP4.5, which is much higher than the average decline over all countries with a voluntary system (34% to 30%). The difference between hard and soft reinsurance market conditions is, on average, in the countries in Figure 5, less than 1% in 2080, which is lower than unaffordability as a percentage of the population shown in Figure 4. This is because, for each modeled household, if the premium becomes unaffordable due to a hard reinsurance market, its decision to not insure will not affect the penetration rate if it did not purchase insurance initially.

Figure 5 shows the penetration rate for countries where flood insurance uptake is voluntary, which is why several EU countries with non-voluntary systems are excluded from this figure. A problem with voluntary insurance systems, as follows from Figure 5, is that uptake declines to such an extent that the majority of the population at risk of flooding is not covered by insurance, and is therefore not financially protected against flood risk. Instead of formal insurance, these households may have to rely on private savings or on government support in the case that they experience damage due to flooding. In Appendix B, we elaborate on the possible extent of required government compensation or private savings to cover uninsured flood damage for EU countries with voluntary insurance systems.

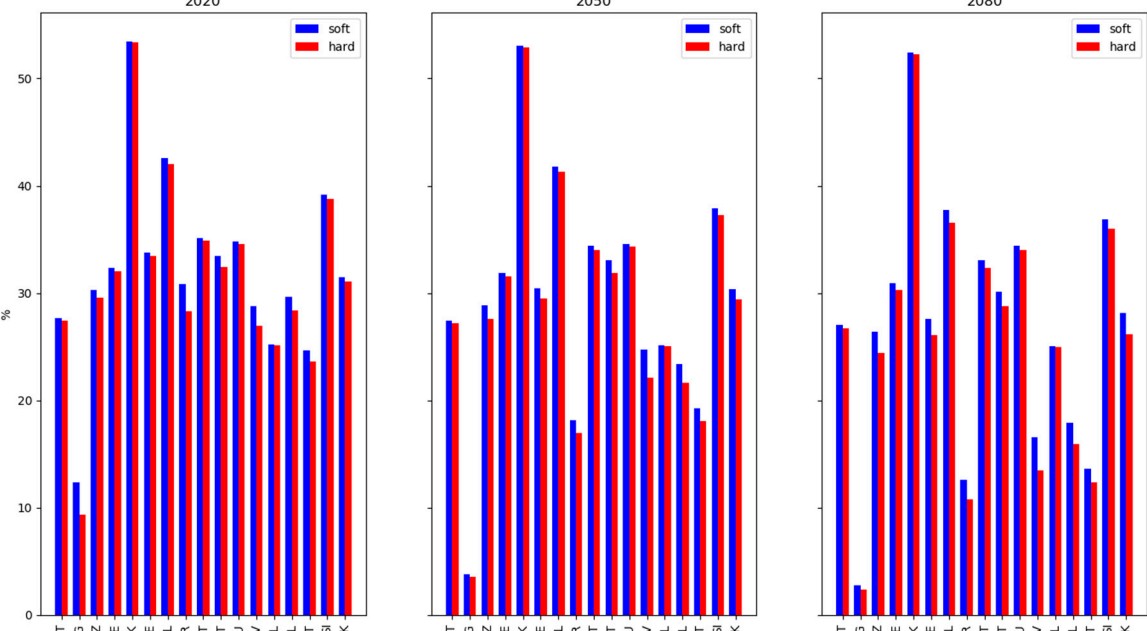

**Figure 5.** Percentage market penetration of voluntary insurance under soft market conditions (blue) and hard conditions (red) for EU countries, in 2020, 2050, and 2080, for RCP4.5. Countries that provide public reinsurance for flood insurance are excluded here, as these are not exposed to global reinsurance market volatility resulting from large natural catastrophes. Additionally, countries with semi-voluntary insurance systems are excluded here, since insurance uptake is a mortgage requirement and is therefore less responsive to capital market shocks.

## 4. Discussion

Volatility on the global capital market for reinsurance, as a result of large catastrophes can have substantial consequences for flood insurance premiums and the unaffordability thereof. We showed the projected development of flood insurance premiums towards 2080 and presented the fluctuation of premiums due to hard and soft capital market conditions for reinsurance, as defined by Kunreuther et al. [11]. Moreover, we plotted the difference in unaffordability and the uptake of flood insurance under hard and soft capital markets for reinsurance and under various scenarios of climate- and socio-economic change. The results of our modeling exercise show a steep increase in insurance premiums towards 2080 and a significant impact of capital market conditions, which is especially evident under the high-end scenario RCP8.5-SSP5. The impact of capital market conditions for the reinsurer can cause considerable changes in the unaffordability of, and the demand for, flood insurance, which can have further repercussions for households that may have to stop insuring and rely on government compensation or private savings in the case that flood damage occurs.

In order to enhance the financial resilience of households to flooding, governments can impose insurance purchase requirements. For example, high penetration rates of flood insurance for households in Sweden (>95%) is a result of uptake being compulsory for household mortgages [46]. Similarly, France attains a penetration rate of 100% by including flood damage coverage in general property insurance and by keeping premiums low for high-risk households due to full risk-sharing amongst its population [46]. However, in the absence of flood risk cross-subsidization, which may have low political support due to the higher financial burden for households in low-risk areas [19], households may not be able to afford the premium. For these households, it may be required by governments to cover the unaffordable share of premiums in the form of subsidies or, as proposed by Kousky and Kunreuther [5], a means-tested voucher, which preserves the incentive for applying risk-reduction measures. From our analysis, it follows that climate change and hard reinsurance capital markets have a relatively small effect on the number of households for whom flood insurance is unaffordable (see

Figure 4). However, we find that the degree of unaffordability increases for households for whom insurance was already unaffordable (see Figure 3), which means that, under a system such as the one proposed by Kousky and Kunreuther [5], a higher financial burden is placed on the government to pay for the subsidy systems.

Moreover, in order to improve the affordability of flood insurance, a government can act as a reinsurer. Governments are generally more risk neutral than private reinsurers, can borrow money on the capital market at lower costs than private (re)insurers, and have alternative funding mechanisms, such as ex post taxation, at their disposal, which makes them less reliant on international capital markets for underwriting flood risk [34]. This type of arrangement is applied in France, where flood risk is covered by private insurers who pass a proportion of the collected premiums to a government backed reinsurer, the CCR (Caisse Centrale de Reassurance), depending on how much it reinsures on the private market [6]. A similar system exists in Spain, where all natural hazards are covered by a publicly backed insurer, the CCS (Consorcio de Compensacion de Seguros) [6]. The recently established Flood-Re in the UK in effect performs as a reinsurance pool for high-risk households in order to facilitate premium cross-subsidization. For every underwritten flood risk, a share of the premium is paid to this fund in order to cover damages for approximately 500,000 households residing in high-risk areas [47]. The underwritten risk for households in lower risk areas are still privately reinsured in the UK.

A noteworthy topic for discussion is that evidence indicates lower reinsurance market volatility resulting from large losses such as natural disasters over recent years [13]. For instance, although 2017 was the most costly year in history in terms of insured damage, reinsurance premiums hardly increased as a result of the recapitalization of (re)insurers [48]. The persistence of a soft market is argued to be due to several factors, including low interest rates, which stimulate capital providers, such as pension funds, to invest in various new forms of reinsurance, such as catastrophe bonds, insurance-linked securities (ILS), or sidecars, which still provide relatively high returns [48]. Besides this, increased capital mobility and flexibility as a result of globalization and technological improvements has largely flattened the cyclic pattern, as money can quickly flow in response to increased returns due to capital shortages [49].

The abundant supply of capital in response to large losses incurred by reinsurers is, however, largely a result of the financial policies of fiscal institutions, which have been maintaining low interest rates since the financial crisis of 2008 [50]. This, in combination with the absence of large catastrophes in the period 2011–2017, led to a high supply of alternative capital [48]. The costly year 2017 did not immediately result in the hardening of the capital market, however, it has been found that the "loss creep", which is the delayed payment of insurance claims due to bureaucratic obstructions, on top of costly disasters in 2018, such as typhoon Jedi in Japan and wildfires in California, are pressuring capital availability for reinsurance companies, which is possibly an end to years of soft reinsurance conditions [48].

It should be noted that, although we portray a scenario of a large natural catastrophe that remotely triggers a hard capital market for reinsurance, in our analysis we do not distinguish between events that may cause this situation. While evidence suggests that large natural disasters are an important driver for capital market conditions and, subsequently, rising reinsurance premiums, other types of events, such as terrorist attacks or economic crises can similarly trigger a hard capital market for reinsurance [13]. Furthermore, in our analysis, we do not explicitly distinguish between remote or local events that cause insurance premiums in the EU to rise. However, the portrayed scenario, where premium setting rules for reinsurers are affected, while those for primary insurers are unchanged, is unlikely to occur after a local catastrophe. For a locally occurring catastrophic event, the primary insurer will also attempt to recapitalize and regain profitability by raising premiums [51], which causes flood insurance premiums to rise by more than what we considered in this study.

In addition to the previous points of discussion, we want to address the interpretation of the presented results, especially concerning the focus areas of this study, which are current 1/100-year

floodplains in EU countries. These areas are expected to experience a high growth in flood risk, which causes problems with the unaffordability and uptake of insurance to be particularly evident there, when premiums are risk-based and uptake is voluntary. Therefore, the results shown in this study represent modeled outcomes for these areas combined per country, and not total national averages.

As climate change is found to increase the frequency and severity of natural catastrophes on a global scale, hard capital markets for reinsurers may occur more often in the future. This study shows that, besides an increasing flood risk within the EU, a hard capital market for reinsurance, as a result of catastrophes that can occur worldwide, can have significant impacts on flood insurance markets within the EU. Previous studies addressing the feasibility of flood insurance in a changing climate have mainly focused on the impact of increasing flood risk within the EU [19,52,53]. This research shows how global interlinkages through the reinsurance market can cause difficulties for EU flood insurance markets, in addition to the more frequently studied impacts of climate change on flood risk in the EU.

## 5. Conclusions

In this study we addressed the impact of climate change on EU flood insurance markets by focusing on the influence of global reinsurance and capital market volatility. Due to a severe catastrophe, or multiple catastrophes in close succession, a high demand for capital by (re)insurers may trigger a "hard" international capital market for reinsurance, where investors raise the price of capital. As a result, reinsurers and, consequently, primary insurers who reinsure on international markets, raise their premiums, due to which the catastrophe can also have an effect on consumers in countries where the catastrophe did not occur.

In this study, we focused on the extent to which flood insurance premiums change in the EU as a result of a global hard capital market compared to a soft market for reinsurers. We find that the impact of a hard market on flood insurance premiums, for EU countries where flood insurance is privately reinsured, becomes higher due to climate change as a result of an increasing average, and variance of, flood risk. Higher premiums cause more unaffordability and a lower market penetration of flood insurance in countries where this is optional. Consequently, a larger uninsured population implies lower financial resilience to possibly larger and more frequent flood events in the future, where households have to rely on private savings, or on uncertain and perhaps partial government compensation, to cover the damage. Dependent on the willingness of governments to provide flood damage compensation, the burden for public budgets may increase significantly, as shown in this study. The additional burden for governments may diminish funds from other welfare enhancing objectives. Moreover, insurance purchase requirements, which are a possible solution to low market penetration, may require large sums of public money to subsidize unaffordability. Introducing a public reinsurer may reduce flood insurance premiums, as governments are able to borrow money at lower rates and are less prone to capital market volatility.

As climate change is expected to increase the frequency and severity of natural catastrophes worldwide, it is likely that hard capital markets as a result of remote natural catastrophes will occur more often in the future. We have projected the impact of this for EU flood insurance markets and proposed potential solutions to improve the performance of flood insurance in the face of climate change. Future research can further elaborate these policy proposals and investigate in more detail possible other indirect impacts of increasing flood risk, such as business interruption and lost tax revenues, as well as policy solutions to limit such adverse effects.

**Author Contributions:** Conceptualization, M.T.; formal analysis, M.T.; Investigation, M.T.; Methodology, M.T., W.J.W.B., and J.C.J.H.A.; supervision, W.J.W.B., and J.C.J.H.A.; writing-original draft, M.T.; writing-review and editing, M.T., W.J.W.B. and J.C.J.H.A.; funding acquisition, W.J.W.B. and J.C.J.H.A. All authors have read and agreed to the published version of the manuscript.

**Funding:** We thank the Netherlands Organization for Scientific Research (NWO), and the EU Horizon 2020-project RECEIPT for funding this research (NWO: VICI grant no. 453.14.006 & VIDI grant no. 452.14.005; RECEIPT grant no. 820712).

**Conflicts of Interest:** The authors declare no conflict of interest.

## Appendix A. Figures RCP8.5-SSP5

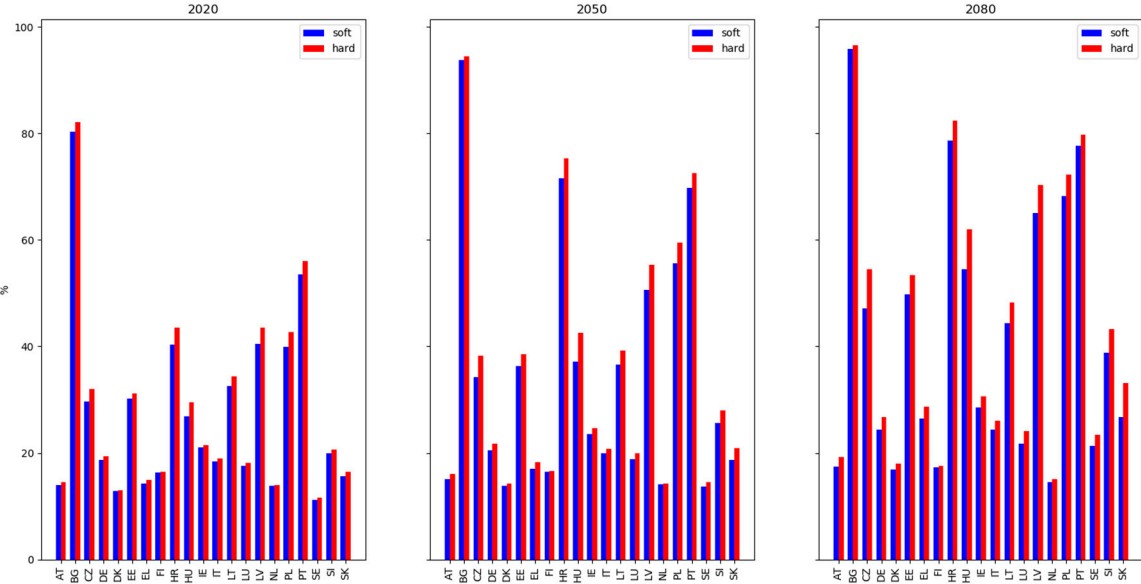

**Figure A1.** The percentage of the population that cannot afford insurance under soft market conditions (blue) and hard conditions (red), for EU countries in 2020, 2050, and 2080, for RCP8.5. Countries that provide public reinsurance for flood insurance are excluded here, as these are not exposed to global reinsurance market volatility resulting from large natural catastrophes.

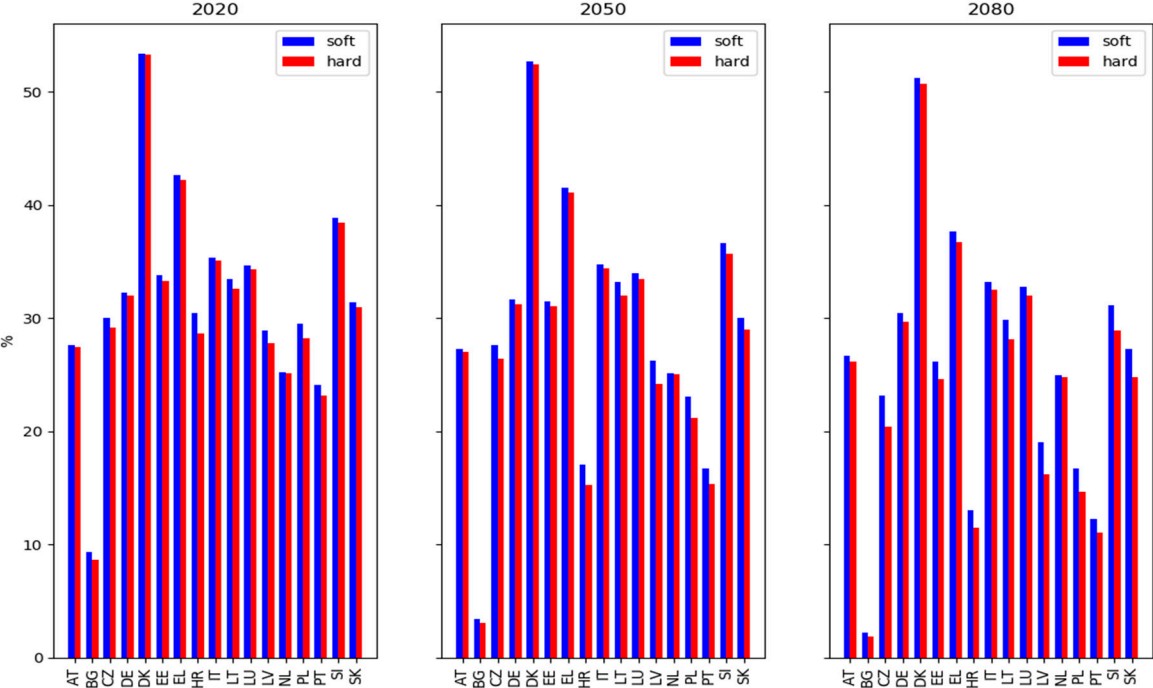

**Figure A2.** Projected uptake of voluntary insurance under soft market conditions (blue) and hard conditions (red), for EU countries in 2020, 2050, and 2080, for RCP8.5. Countries that provide public reinsurance for flood insurance are excluded here, as these are not exposed to global reinsurance market volatility resulting from large natural catastrophes. Additionally, countries with semi-voluntary insurance systems are excluded here, since insurance uptake is a mortgage requirement and is therefore less responsive to capital market shocks.

### Appendix B. Uninsured Annual Expected Damage (Potential Burden for Governments)

After computing the insurance penetration rate and the risk of flooding per household, which is determined by dividing the expected annual damage by the number of households per region, we are able to estimate the annual uninsured risk. This is calculated by multiplying the inverse of the penetration rate by the number of households per region, which is then multiplied by the average flood risk per household. The resulting figure expresses the amount of annual risk that is not covered by formal insurance, which will need to be funded either by governments or private savings if the damage is to be fully recovered from.

Figure 1, below, expresses the uninsured yearly expected damage for the two climate scenarios for three time steps under soft reinsurance market conditions. Figure 2 illustrates the same under hard market conditions. The difference between the two market conditions is highest for RCP8.5 in 2080, where the uninsured risk over all countries is 1.5% higher in a hard compared to a soft market state. More significant is the impact of climate change over time, which is the reason we chose to compare the climate change scenarios instead of the capital market conditions within the graphs. Under soft market conditions the differences between RCP4.5 and RCP8.5 are 22%, 109%, 227% for 2020, 2050, and 2080, respectively, taken over all countries combined. However, it can be seen that several countries show high estimates of uninsured damage. These countries similarly show high premiums and low levels of uptake in Figure 2; Figure 5, which is partly the cause of the high degree of uninsured risk. The three countries showing the highest results—Germany (DE), Poland (PL), and Portugal (PT)—in a soft reinsurance market have on average approximately €900 million uninsured risk in 2020 under RCP4.5 and €1.1 billion under RCP8.5, which is expected to rise to €2.4 billion and €5 billion in 2050, and in 2080 it is estimated at €5.6 billion and €19 billion.

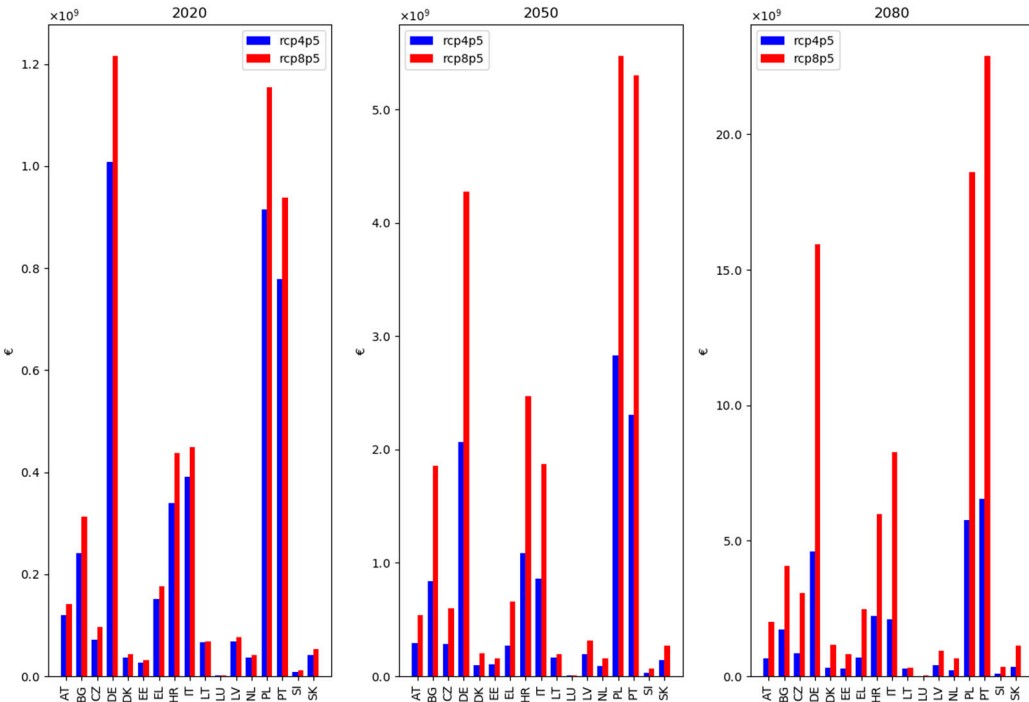

**Figure A3.** Uninsured annual flood risk for EU countries with voluntary insurance systems, under soft reinsurance market conditions, for RCP4.5 (blue) and RCP8.5 (red), for 2020, 2050, and 2080. Countries that provide public reinsurance for flood insurance are excluded here, as these are not exposed to global reinsurance market volatility resulting from large natural catastrophes. Additionally, countries with semi-voluntary insurance systems are excluded here, since insurance uptake is a mortgage requirement and is therefore less responsive to capital market shocks.

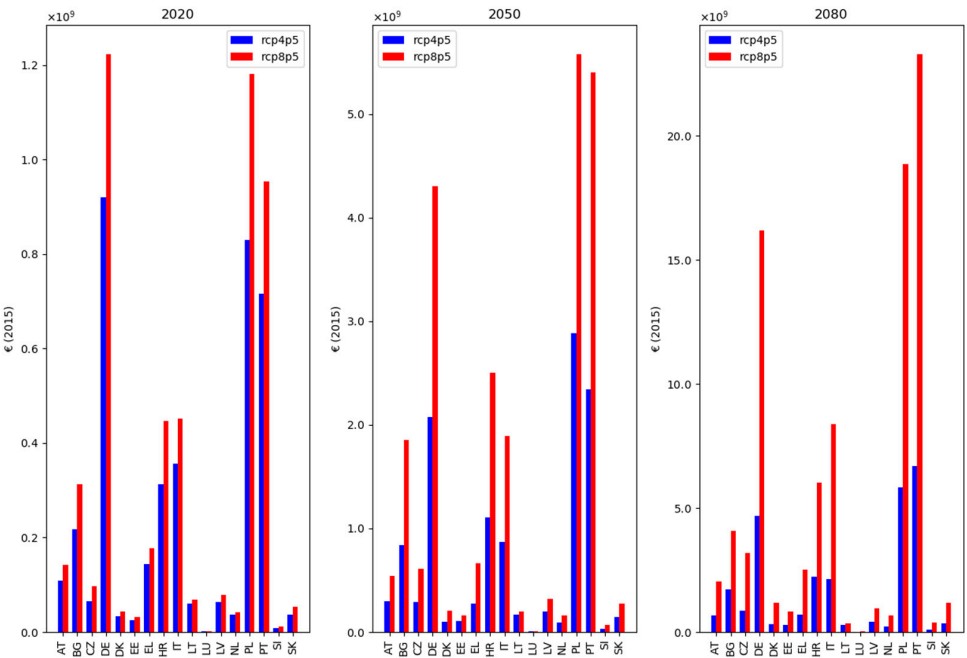

**Figure A4.** Uninsured annual flood risk for EU countries with voluntary insurance systems, under hard reinsurance market conditions, for RCP4.5 (blue) and RCP8.5 (red), for 2020, 2050, and 2080. Countries that provide public reinsurance for flood insurance are excluded here, as these are not exposed to global reinsurance market volatility resulting from large natural catastrophes. Additionally, countries with semi-voluntary insurance systems are excluded here, since insurance uptake is a mortgage requirement and is therefore less responsive to capital market shocks.

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
