# Peer review of "Impacts of Climate Change and Remote Natural Catastrophes on EU Flood Insurance Markets: An Analysis of Soft and Hard Reinsurance Markets for Flood Coverage"

_atmosphere, doi:10.3390/atmos11020146_

Round 1

Reviewer 1 Report

Why do the number of countries in the graphs decline from 26 (Fig. 2) to 17 (Fig. 5). This appears to be important (e.g. the UK is missing after Figure 2) but is not explained. How would this affect the conclusions? No longer truly "EU" if only 60% of countries are included? Does it not matter that insurance and reinsurance companies have extensive non-flood portfolios (life; motor; etc)?  Thus risk is spread much further than the paper assumes. Could the likely costs to governments under the different scenarios be assessed?  The word "remote" in the title is ambiguous. Do you mean "rare" or "extreme"? I think you do.

Reviewer 2 Report

An interesting paper about a relevant issue with an appropriate modeling approach that gives important insights. However, this analysis, while important, can only be seen as a first step into this area and therefore should include some more discussion on the limitations of this approach as well. This should not be too difficult as this is related to the discussion and conclusion section. I also would change the title as the focus is on hard and soft markets. While the reason for that is explained and discussed in regards to remote natural catastrophes which is important it does not provide the clear link to actual remote effects (this is also stated in the text, e.g. page 13 line 475, but is actually not the case, hence it needs some revisions here what was actually shown in the analysis; I think line 480-481 states the hypothesis tested). Hard and soft markets could also actually happen due to other reasons as well. This should be a little bit elaborated in the discussion section as well as ways forward. The focus on the 1/100 year flood zones is also sufficiently explained, however, also something which limits the general conclusion. What is with low probability/high impact events, due to their huge losses there could be large indirect effects (e.g. indebtedness, opportunity costs) which would be needed and should  be addressed (at least in future work). Using an subjective utility maximization approach is also defendable, however, risk aversion may be not due to only the expectation but also the spread of risk, e.g. variance or tails of the risk distribution (Value at risk). This should also be discussed a little bit in the limitations section and outlook.

Round 2

Reviewer 1 Report

No comments